# Structural Relationships among Strategic Experiential Modules, Motivation, Serious Leisure, Satisfaction and Quality of Life in Bicycle Tourism

**DOI:** 10.3390/ijerph182312731

**Published:** 2021-12-02

**Authors:** Rui Guo, Xiaoying Liu, Hakjun Song

**Affiliations:** 1Department of Leisure, Service and Sports, Pai Chai University, Daejeon 35345, Korea; guoruippx@gmail.com; 2Department of Tourism Management, Pai Chai University, Daejeon 35345, Korea; lll12231108@163.com; 3Department of Hotel and Service Management, Pai Chai University, Daejeon 35345, Korea

**Keywords:** bicycle tourism, motivation, serious leisure, satisfaction, quality of life, structural equation modeling, Qinghai Lake, China

## Abstract

It seems that people’s quality of life can be positively influenced through bicycle tourism. Bicycle tourism can be an effective measure to enhance serious leisure, tourism satisfaction, and quality of life. To verify this empirically, a survey was conducted of bicycle tourists who visited Qinghai Lake in China during an international road bike race. The purpose of the present research is to prove the association between latent variables related to bicycle tourism through statistical analysis. For this, hypothetical relationships based on tourism motivation, serious leisure, tourism satisfaction, and quality of life were presented as research models. As a result of empirical analysis, it was analyzed that friends and nature had an effect on serious leisure among the motivation of bicycle tourism. In addition, it was found that the level of serious leisure for bicycle tourism exerted a positive influence on the satisfaction and quality of life. This suggests that bicycle tourism can improve the quality of life during travel to Qinghai lake by bicycle and revealed the crucial role in relationships is serious leisure.

## 1. Introduction

As a new type of tourism and one of the most personalized tourism activities, bicycle tourism is attracting more and more people. Compared to traditional mass tourism, bicycle tourism is recognized as a more free, flexible and in-depth tourism experience. In general, tourists can also satisfy their aesthetic desires and gain friendship and deviance enjoyment through tourism behaviors. From this point of view, bicycle tourism can be considered to be one of the various tourism actions that can sufficiently satisfy the needs of tourists and can satisfy a significant portion of the tourism demands of people in the modern society. In relation to bicycle tourism, an international road bike race has been held in Qinghai Lake, China, and this competition has generated various economic and social benefits to the host area through bicycle tourism. However, despite the contribution of such international road bike races, studies to understand the behavior of bicycle-related tourists through proper theories have not been conducted often. With the increasing desire of people for quality tourism activities, interest in bicycle tourism is gradually increasing, which can enhance quality of life via meaningful activity. Currently, the insufficient researches are either isolated qualitation ethnography or quantitation analysis which are primarily depictive and on the foundation of small and separated specimens [1,2]. For example, Cater [1] stressed that more bicycle tourism researches based on theories are required. Zhang [3] suggested policy, principles, and monitoring and evaluation processes as a strategy for sustainable development of the sporting event for athletes participating in the Tour of Qinghai Lake cycling race through IPA analysis. Based on this, present study attempts to address this need through strategic experiential modules (SEM) of the relationships between serious leisure, motivations, satisfaction and quality of life, across Qinghai Lake sample of bicycle tourists. One of the research goals of this study is to obtain insights by confirming and expanding the motivation of bicycle tourism as an example. In order to achieve the research objectives, the present research will reveal the relationship between the main motivations of bicycle tourism (i.e., friends, nature, and deviation), serious leisure, tourism contentment, and life quality. Based on this empirical analysis, the present research can provide theory-wise and application-level implications for the development of bicycle tourism. In addition, this study is expected to provide an opportunity to promote academic development by expanding knowledge on the interrelationship between tourism motivation, tourism satisfaction and quality of life from a long-term perspective.

## 2. Literature Review

### 2.1. Bicycle Tourism and Tourism Motivation

Since the late 18th century, the invention and improvement of bicycles have gone evolved over 200 years. Simonsen, Jorensen, and Robbins [4] defined bicycle tourists as tourists who use bicycles as their main transportation device during travel. Ritchie [5] defined bicycle tourism is any activity undertaken by people on vacation for longer than 24 h or one night and for whom the bicycle is an integral part of this trip. Ritchie [5] categorized cycling motivation into seven categories: autonomy and achievement, solitude, exploration, physical challenge, search for similarities/avoid similarities, social interaction, and escape from society. Commercial activities may not be the main purpose of bicycle tourists, but it is true that bicycle tourism brings millions of pounds of infrastructure construction costs to the local area [5]. Kormosne Koch [6] believes that bicycle travel is not only a leisure activity, but also an environmentally friendly and a highly practicable means of transportation. The definition of bicycle tourists directly influences the definition of bicycle tourism [7]. The main motivation for cycling shown in Beierle [8] study includes lifelong dreams, sense of achievement and pursuit of integrity, enhancing family/friends opportunities to get along, health reasons, career changes, relationship changes, memorial or celebration, and leisure or adventure. Downward and Lumsdon [9] also showed that cyclists are more likely to socialize with their friends taking breaks and chatting together than other types of travelers.

Motivation is an important variate in bicycle tour research and serves as the starting point in researching tourists’ behaviors and exploring tourists’ consumption. There are 3 commonly utilized theory structures to elucidate tourists’ motivations. Firstly, Maslow’s [10] hierarchy of needs model has influenced substantial tourism intention researches [11]. Secondly, Iso-Ahola’s [12] popular dichotomous model (pursuing/running away, individual/inter-personal) has offered a new theory dimension for tourism motivation researches. Additionally, the academic world has been attempting to reveal tourists’ intentions via the concept of imbalance [13], which concludes that mankind is prone to maintain the balance status in which no clashes exist between anticipation and reality. Disturbing the balance in such a theory structure drives people to act accordingly. Thus, this paper chooses nature (NAT), deviance (DEV) and friends (FRI) as three core elements present in the motivation of bicycle tourists.

### 2.2. Serious Leisure (SL)

Stebbins [14] argues that the serious leisure (hereafter SL) is a “truly integrated theoretic perspective”, a “formal grounded theory” [14], an “established theory” [15], and “a valid and useful explanation of human motivation, group formation, collective action, and the like” [14]. Blackshaw [16] reveals that SL has taken researches on leisure “in a new direction from other more conventional approaches which largely tend to focus their critical gaze on the dichotomy between work and leisure.” Gillespie, Lefler, and Lerner [17] hold that the application of the SLP “heralded a conceptual shift in how leisure was studied.” Worthington’s [18] view is that, with the occurrence of SL, “the very idea of ‘leisure’ was turned upside down,” whereas, for Dilley and Scraton [19], it was a “significant theoretical development.” Even commentators suggesting massive variations in the perspective identify its theory-wise position.

However, for masses of people, leisure activities such as bike tours are vital, and encompass nature, friends and deviance. The normal definition of SL is that it’s the all-round seeking of amateur or voluntary activities that are adequate and intriguing for participants to discover a profession which acquires and expresses an integration of their specific abilities and experiences’ [20]. The essence of SL could be displayed via comparing it to what Stebbins calls “casual leisure”, or leisure consisting of “immediate, intrinsic comparatively transient intriguing activities that require almost no specific training for the purpose of enjoying them” [21].

Nevertheless, it can be argued that the idea of SL has been almost neglected in researches pertaining to leisure [22], and practically utterly neglected in studies on tourism [23]. Maybe such phenomenon is due to what Stebbins points out that ‘At first glance, the combination of tourism and SL is incongruous for the majority of people, just such as the inappropriate combination of ice creams and pickles, which the majority of us wouldn’t even care to give it a fair thought’ [23]. Both leisure and tourism are related to activities which are usually on the foundation of the seeking of ‘happiness’ [23]. Nonetheless, as will be displayed, SL offers a helpful theory-wise concept to depict and elucidate some bike tour activities.

In the past relevant leisure research, it is found that motivation and SL are related [24]. Herman [25] studied the relationship between cyclists’ leisure motivation and SL, and found that there was a correlation between them, among which demographic variables had a high effect. The entire SL qualities assessed, apart from the unique ethos, were discovered to be associated with previous experiences and centrality-to-life style in a positive way [26]. Tsai [27] studied the relationship among motivation and SL of leisure tennis players. The results showed that participation motivation had a significant positive effect on SL, indicating that the higher the participants’ recreational motivation, the higher the SL. It was also revealed that participants’ recreational motivation and SL have a significant effect on recreational specialization. Hence, the present research puts forward the assumption as follows:

**Hypothesis** **1** **(H1).**
*The FRI is important sources of SL.*


**Hypothesis** **2** **(H2).**
*The DEV is important sources of SL.*


**Hypothesis** **3** **(H3).**
*The NAT is important sources of SL.*


### 2.3. Satisfaction (SAT)

In the majority of customer behavior models, satisfaction (hereafter, SAT) serves as the center between the antecedents and descendants’ variates. A variety of customer behavior models have verified the roles of SAT in tourists’ loyal status [28,29]. The former takes SAT as processes of assessment that the experience is as enjoyable as expected [30]. The present research uses the result viewpoint of tourism SAT, coinciding with substantial scholars suggesting that tourism SAT is the emotional status originated from a positive assessment of the tourist’s experiences [31,32]. Despite the fact that the processes of forming SAT include cognition evaluations of stimuli, the idea of SAT is thought to be associated with people’s motivations. In addition, scholars have revealed that SAT is context-specific concept relying on the interesting characteristics of commodities or services [31]. This applies to two different kinds of SAT: transaction-specific SAT and general SAT [33]. The former denotes a direct emotional response to a separate service experience, whereas the latter denotes the general psychology status on the foundation of the whole tourism experiences [34]. Tourists might have dissatisfaction when they encounter services such as flight postponement (generating low SAT with the specific transaction), but remain content with the entire tour (namely, general SAT) due to the rest of the enjoyable services. For that reason, general SAT is the integration of the entire past satisfactory experiences based on specific transactions [35]. General SAT has been considered a more steady concept in contrast to satisfactions based on specific transactions and was more emphasized in the previous decade [31].

In the studies on sightseeing, massive past researches defined tourism SAT at the general level as a cumulative psychology status stemmed from the tour experiences [32,36,37]. Such a concept of tourism SAT has been broadly used to explore the association between SAT and the rest of concepts such as tourists’ loyal status and revisiting motivation [32,38,39,40]. Thus, this study proposes the second hypothesis as follows:

**Hypothesis** **4** **(H4).**
*SL casts a positive influence on SAT.*


### 2.4. Quality of Life (QoL)

The concept of quality of life (hereafter QoL) is implicit in many researches on sightseeing influences. As per the OECD [41], QoL can be expressed as “the notion of human welfare (well-being) measured by social indicators rather than by ‘quantitative’ measures of income and production”. QoL usually equals subjective welfare or life satisfaction [42,43]. The construct of QoL is related to the understanding of an individual’s perceived SAT with the environment where he lives. Bonsjak et al. [44] put forward a model to identify the QoL which can be produced by consumers during sightseeing, where 7 standards can be generated by tourism destinations to yield the QoL for consumers. Those standards involve self-congruent, functionally congruent, comfort-wise congruent, healthy and safe, morally congruent, economically congruent, and hedonic aspects. Neal et al. [45] hold that QoL and SAT have a tight association requiring an all-round modeling. Despite the fact that the present research won’t review the broad debates on defining QoL and the associated constructs, the definition of OECD is essential for the discussions herein as it highlights the value of things instead of revenue and productivity. Since the consideration of economy results associated with revenue and productivity has been predominant in massive social representations of sightseeing, QoL provides the possibility to expand the explorations of sightseeing results. On the foundation of the preceding reasoning, the third assumptions put forward in the present research are stated below:

**Hypothesis** **5** **(H5).**
*SAT casts a positive influence on QoL.*


## 3. Methodology

### 3.1. Measurement Items and Survey Questionnaire

This research designed a self-reported questionnaire for exploring the association between motivation, SL, SAT, and QoL. The self-reported questionnaire comprised 5 parts: First, three dimensions (nature, deviance, and friends) with 9 indicators to measure the motivation were adapted from studies of [28,46,47,48]. Secondly, SL section had 5 items adopted from Stebbins [49] and was developed to measure tourists’ SL with bicycle tourism around Qinghai Lake. Thirdly, SAT was assessed with 4 items from [34]. Fourthly, the QoL section had 3 indicators borrowed from [50] to evaluate the QoL of tourists. Finally, the demography profile part comprised gender, education level, profession, the state of matrimony, monthly income, residence, and age. In the initial 4 parts, the entire indices were identified by using a 5-point Likert-type scale that ranged from 1 = ”strong disagreement” to 5 = ”strong agreement” (see Table A1). This research questionnaire includes (1) demographic characteristics of bicycle travel, (2) bicycle motivation, (3) SL, (4) SAT, and (5) QoL. Questions asking about research motivation are based on [28,46,47,48]. Stebbins [49] believed that being active and seeking participatory experiences, although necessary for tourism of special interest, are imperative but inadequate conditions of its sub-type, culture-related sightseeing. Thus, SL is a key element for tourists’ motivation and friends, nature, and deviance as motivation of tourists’ motivation were present in the survey [51]. Finally, satisfactions based on specific transactions denote a direct effective response to a separate service experience, whereas general SAT denotes the general psychology status on the foundation of the whole tour experiences [34]. Bonsjak et al. [44] put forward a model to identify the QoL which could be produced by consumers during sightseeing. Thus, there also have questions in the survey about tourist’s SAT and QoL [50].

### 3.2. Data Collection and Demographic Profile of Samples

In this study, five sites around Qinghai Lake were selected for investigation, including the Dayu tribe, Qibingying, the Erlang sword, the Heima River, and Birds Island. The peak season of Qinghai Tourism resulted in a large number of people who participated in bicycle tourism, and this makes the sampling more random and representative. Therefore, the target population of this study were bicycle tourists form China who traveled in the above 5 sites. By virtue of stochastic sample collection, the data were harvested from July 15th, 2018, to August 15th, 2018, before the COVID-19 pandemic outbreak. Finally, 400 questionnaires were used and overall 300 finished were utilized for further data analyses except ineligible ones (interviewees giving the identical assessment points to the entire indexes, interviewees who were hasty, or incomplete questionnaires).

In this study, R-statistic was applied to perform descriptive statistics at first. On the foundation of Anderson and Gerbing’s recommendations, this research completed the SEM via a two-step method [52,53]. Moreover, for the purpose of ensuring inner consistency as well as construct validity and dependability, CFA was employed to test the modeling for the entire variates. Finally, the structural relationships between the four variables were verified by SEM. Figure 1 is the proposed conceptual model.

## 4. Results

### 4.1. Descriptive Statistics

The result of Table 1 displays the demography features of the respondents. The sample was comprised of more males (73%) than females (27%). The participants ranging from 20 to 29 years old predominated the specimen, taking up 43%, followed by the people between 40 and 49 (20.4%). The majority of the interviewees finished tertiary education, such as junior colleges (33.7%) or universities (35%). Their income level was rather low, with 30.3% of the respondents earning a monthly income of less than 3000 ¥, and 28% for 3000–4999 ¥. Regarding marriage, the proportion of the respondents who were married is (47.7%), not married is (47.7%), and other is (4.6%). Staff (31%), students (22.3%) and self-employees (11.3%) were predominant occupations of the residents.

### 4.2. Measurement Model

Firstly, the CFA was finished via R-studio with a MLE to identify the discriminant validity, and convergent validity of the indices. The fitting indexes reveal that the exterior model fitting was satisfactory with χ^2^ =289.396, df = 155, χ^2^/df = 1.867, *p* < 0.001, CFI = 0.960, NFI = 0.918, TLI = 0.951, and RMSEA = 0.054 [53]. Convergent validity was evaluated using CR and AVE scores [54]. As presented by Table 2, all indicators exhibited AVE of over 0.5 and CR of above 0.7, revealing a satisfactory convergent validity and inner consistency [55]. In addition, the values of AVE over 0.5 reveal potent convergent validity, which suggests that over 50% variations in a specific construct is elucidated by the specific indices [56]. To be specific, for all constructs within Table 2, the square root of the AVE was higher in contrast to every coefficient of association for discriminant validity [57] and Cronbach’s alpha was above 0.7 for dependability of inner consistency [58]. Finally, finding out the value of the confidence interval between the highest FRI correlation coefficient and the highest QoL correlation coefficient in the correlation coefficient of the latent variable. According to the correlation (coefficient ± 2 × standard error and 0.791 ± 2 × 0.037), the values obtained are 0.865 and 0.717, without 1. Therefore, the value of discriminant validity obtained using the correlation coefficient of the latent variable is satisfied [59].

### 4.3. Structural Model

This section presents the results of structure modeling depicting bicycle tourism’ decision-making processes. The proposed theoretical model will be analyzed, and hypotheses will be tested in this section by employing R-studio [60]. The structure model was found to match the data well with better fitting accuracy (χ^2^ = 373.301, df = 162, χ^2^/df = 2.304, *p* < 0.001, CFI = 0.937, NFI= 0.895, TLI = 0.926, RMSEA = 0.066. The modeling exhibited a sufficient fitting to the data. Of the five proposed hypotheses except H2 rejected all hypotheses were accepted. As shown in Table 3, H1: FRI casts a remarkable influence on SL (β = 0.748, t result = 8.33, *p* < 0.001), H2: DVE has on significant influence on SL (β = −0.103, t result = −0.948, *p* > 0.05). H3: NAT casts a remarkable influence on SL (β = 0.263, t result = 2.673, *p* < 0.01), H4: SL casts a remarkable influence on SAT (β = 0.992, t result = 82.664, *p* < 0.001), H5: SAT casts a remarkable influence on QL (β = 0.909, t result = 45.098, *p* < 0.001). Indicating that greater scores on FRI and DEV meant higher scores on SL. Moreover, SL was found to positively correlate with SAT, SAT was found to positively correlate with QoL. Meaning that higher scores on SL meant higher scores on SAT, higher scores on SAT meant higher scores on QoL. However, the association between DEV to SL was not significant, *p* > 0.05 implying that there was lack of evidence for demonstrating a relationship between DEV to SL. The assumption test outcomes are presented by Table 3. Figure 2 is the structural model results.

## 5. Discussion

This study used the research methodology of SEM to test the causality between the motivation factors of bicycle tourism, SL, bicycle tourism SAT, and QoL. As a result of empirical analysis, it was found that FRI and NAT were main motivating factors determining the level of SL, but DEV was not. The results of current study revealed that SL is closely related to the motivation of Qinghai lake bicycle tourists is consistent with the assertions of previous studies [61]. The outcomes of the present research and past researches show that the level of an individual’s SL for bicycle tourism is determined by the degree of motivation of bicycle tourists. In addition, the analysis result which SL is related to SAT and SAT increases QoL indicates that bicycle tourists are satisfied through SL, which is deeply related to improvement of QoL.

The present research has yielded certain theory-wise significance. This study supplemented the findings of previous studies related to QoL and SL by revealing that tourism satisfaction realizes the mediation of the association between SL and QoL. In the meantime, several researchers have argued that SL is an important predictor of QoL. According to previous studies, SL can positively predict QoL, and related indicators were found to be education level, self-development, self-expression, and self-satisfaction (pleasure) [62]. In another study, SL (e.g., career contingency and personal effort) was analyzed to cast a positive effect on the QoL of taekwondo participants [63]. Chen [64] argued that a close association exists between SL and QoL for 264 elderly volunteers in Taiwan, which might alter according to the degree of support for spouses. In this regard, this study suggests a new perspective that SL is vital for SAT and QoL. In other words, the QoL is affected by the SAT formed through SL. In this respect, the results of this study have high academic value in that they examine the relationship between the motivation of nature, friends, and deviance, SL, SAT, and QoL from various perspectives. The analysis outcomes of the present research examine the influence association between QoL and other related variables, and based on this, it can provide various implications for bicycle tourism and the development of tourism destination where bicycle tourism is performed.

Some practical implications can be gained from this study. First, for practical insights, it is important to meet the recreational needs and life expectancy of bicycle tourists in Qinghai Lake. Specifically, by increasing social opportunities through bicycle tourism and providing many health-related, nature-friendly places, interest in bicycle tourism is raised, increasing SAT with bicycle tourism itself, and improving the QoL through bicycle tour. As such, it’s imperative to elevate the re-visit intention of bicycle tourists and to achieve the development of local tourism through bicycle tourism. In Qinghai Province, despite the continuous growth of bicycle tourism over the past decade, development plans for bicycle tourists have not materialized in relation to bicycle routes. To this end, Qinghai Province needs to actively build bicycle tourism infrastructure, such as developing bicycle routes to meet the current demand for bicycle tourists, increase bicycle tourism-related activities and promote economic development. Since bicycle tourists can spend more and stay longer in tourist destinations than regular visitors, the construction of bicycle tourism infrastructure will affect bicycle rental and repair shops, accommodation, food and beverage services, and development of key areas related to bicycle road development. In summary, this paper suggests that Qinghai should pursue bicycle tourism as a possible economic growth point and improve the prospects of regional development. Due to the diverse nature and motivating factors of friends, bicycle tourism can provide a differentiated tourism experience as a more environmentally friendly tourist transport and an opportunity to run a business in local development.

## 6. Conclusions

Finding a means to revitalize the local economy has been recognized as a very important issue due to the changing economic and social conditions of the region, such as a decrease in local industrial activity, unemployment and an elevation in the quantity of workers moving to cities [65]. Tourism has served as the most powerful tool among alternatives to develop regional economic development in rural and surrounding areas. Among the various tourism businesses, bicycle tourism, which can be operated in a sustainable form in both supply and demand aspects, will need to be particularly emphasized. Recently, with the rise of the concept of low-carbon environmental protection and life, bicycles have gradually been combined with tourism activities as a representative green transportation method. Such bicycle tourism has established itself as one of the new fashions and trends in the tourism field. This study, conducted on bicycle tourists visiting Qinghai lake, provided an opportunity to empirically show that bicycle tourism can actually contribute to the improvement of the QoL of consumers and industrial aspects. Based on these results, it can be concluded that SL should be recognized as an important issue in relation to QoL.

In addition, this study focuses on the data and findings predate the pandemic outbreak, but COVID-19 has been a before and after in the tourism industry worldwide. Likewise, there are four limitations in this paper. Firstly, although friends, deviance and nature are main constrain factor for tourism motivation, there are still some crossed factors which may also affect tourism motivation and tourist’s satisfaction, such as health, values, and so on. Therefore, These crossed factors can be taken into consideration in future studies. Secondly, there is limitation of language bias as the questionnaire was only prepared into Chinese. As Mainland China are also main resource area of Qinghai Lake bicycle tourism, but Qinghai lake is also famous in Japan. However, Japanese who come to Qinghai lake cannot read Chinese very well. Therefore, in order to get generalized respondents, questionnaires in various languages should be made, which may also enhance the difficulty in conducting survey. Thirdly, the sample size in the current study is not balance, with more questionnaires were spread to Mainland Chinese tourists, which do not satisfy the requirements of multi-group test. In further studies, a variety of survey methods can be used to equalize the sample size of eastern and western tourists. It will be a good trial for future research to compare the difference between western tourists and eastern tourists in order to understand Relationships among Motivation, SL, SAT and QoL in bicycle tourism. Finally, the non-significance relationship between motivation and QoL differed from previous literature [26,66,67]. The difference may stem from the problem of multicollinearity of measurement. Additionally, the difference may originate from the different contexts for the studies being conducted. Future studies should further investigate the relationship between motivation to bicycle tourists and QoL, and the bicycle tourism questions on the most recent one should reflect general bicycle tourism behavior.

## Figures and Tables

**Figure 1 ijerph-18-12731-f001:**
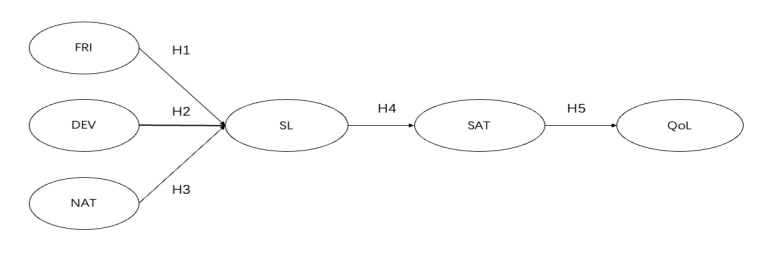
Proposed conceptual model. Notes: FRI = friends, DEV = deviance, NAT = nature, SL = serious leisure, SAT = satisfaction, QoL = quality of life.

**Figure 2 ijerph-18-12731-f002:**
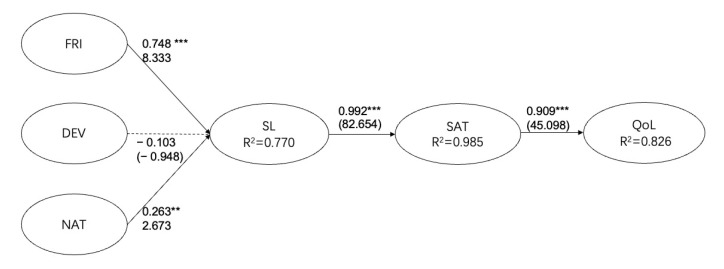
Structural model results. Notes 1: fitting accuracy statistic results: = χ^2^ = 373.301, df = 162, χ^2^/df = 2.304, *p* < 0.001, CFI = 0.937, NFI = 0.895, TLI = 0.926, RMSEA = 0.066. Notes 2: FRI = friends, DEV = deviance, NAT = nature, SL = serious leisure, SAT = satisfaction, QoL = quality of life. Notes 3: ** *p* < 0.01, *** *p* < 0.001.

**Table 1 ijerph-18-12731-t001:** Respondents’ demographic characteristics (*n* = 300).

Feature	*N* (%)
Sex	
Man	219 (73%)
Woman	81 (27%)
Educational background	
Lower than upper secondary education	72 (24%)
Junior colleges	101 (33.7%)
Universities/colleges	105 (35%)
Graduate schools	22 (7.3%)
Profession	
Student	67 (22.3%)
Staff employee	93 (31%)
Worker	26 (8.7%)
Civil servant	17 (5.7%)
Self-employed	34 (11.3%)
Researcher	18 (6%)
Education employee	17 (5.7%)
Other	28 (9.3%)
Marital status	
Not married	143 (47.7%)
Married	143 (47.7%)
Others	14 (4.6%)
Monthly income level	
Less than 3000 ¥	91 (30.3%)
3000–4999 ¥	84 (28%)
5000–6999 ¥	79 (26.3)
7000–8999 ¥	18 (6%)
9000 ¥ or more	28 (9.4)
Residence	
Beijing	104 (21.2)
Shanghai	122 (24.9)
Guangzhou	134 (27.4)
Shenzhen	130 (26.5)
Age	
Less than 20	40 (13.3%)
20–29	129 (43%)
30–39	40 (13.3)
40–49	61 (20.4)
Over 50	30 (10%)

**Table 2 ijerph-18-12731-t002:** Results of measurement model.

Constructs	FRI	DVE	NAT	SL	SAT	QoL	Items	Standardized Factor Loading
Friends: (FRI)	**0.640**						FRI 1	0.836
							FRI 2	0.784
							FRI 3	0.798
Deviance: (DVE)	0.629	**0.703**					DVE 1	0.807
							DVE 2	0.834
							DVE 3	0.872
Nature: (NAT)	0.58	0.744	**0.719**				NAT 1	0.885
							NAT 2	0.851
							NAT 3	0.807
Serious Leisure: (SL)	0.636	0.364	0.428	**0.559**			SL 1	0.770
							SL 2	0.776
							SL 3	0.702
							SL 4	0.782
							SL 5	0.705
Satisfaction: (SAT)	0.644	0.482	0.501	0.78	**0.669**		SAT 1	0.863
							SAT 2	0.858
							SAT 3	0.725
Quality of Life: (QoL)	0.791 *	0.626	0.605	0.731	0.675	**0.736**	QoL 1	0.860
							QoL 2	0.842
							QoL 3	0.872
CR Cronbach’s alpha	0.842 0.847	0.876 0.874	0.885 0.883	0.864 0.862	0.858 0.845	0.893 0.892	Model fit S-B χ^2^(df): 289.396 (155) Normed S-B χ^2^: 1.867 *p* < 0.001 CFI: 0.960 NFI: 0.918 TLI: 0.951 RMSEA: 0.054	

Notes: All factor loadings have significance when *p* < 0.001; The entire bold-faced diagonal elements that appear in the correlation of the construct’s matrix denote the square roots of AVEs. *: The highest value of related variables among latent variables.

**Table 3 ijerph-18-12731-t003:** Normalized parametric estimation of structure model.

Assumptions		Coefficients	t-Value	Testing of Assumptions
H1	FRI → SL	0.748 ***	8.333	Accepted
H2	DEV → SL	−0.103	−0.948	Rejected
H3	NAT → SL	0.263 **	2.673	Accepted
H4	SL → SAT	0.992 ***	82.654	Accepted
H5	SAT → QoL	0.909 ***	45.098	Accepted

Notes: ** *p* < 0.01, *** *p* < 0.001.

## Data Availability

The dataset used in this research are available upon request from the corresponding author. The data are not publicly available due to restrictions i.e., privacy or ethical.

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
