# Peer review of "Structural Relationships among Strategic Experiential Modules, Motivation, Serious Leisure, Satisfaction and Quality of Life in Bicycle Tourism"

_ijerph, 2021, doi:10.3390/ijerph182312731_

Round 1

Reviewer 1 Report

Thank you for the possibility to review your research. But there some little o errors throughout the manuscript.

The Literature Review is too long.

For the statistical analysis, more details and processing approaches are hopefully to be seen, as improving the replication of this study.

In the discussion section, please do not repeat results section while summarise research findings.

I think you should add the Conclusion part. Please revise it.

Author Response

We appreciate the opportunity to revise and resubmit our paper titled “Structural Relationships among Strategic Experiential Modules, Motivation, Serious Leisure, Satisfaction and Quality of Life in Bicycle Tourism.” We are grateful for the time you put forth for your critiques and revision recommendations. Your vital comments and suggestions have contributed substantially to improving the presentation and overall quality of our study. The corresponding changes and refinements made in the revised paper are summarized in our responses below.

Thank you for the possibility to review your research. But there some little o errors throughout the manuscript. The Literature Review is too long.

Response: We appreciate your valuable comments. As suggested, we have modified the Literature Review section

For the statistical analysis, more details and processing approaches are hopefully to be seen, as improving the replication of this study.

Response: We appreciate your valuable comments. As suggested, “more details and processing approaches”, this study followed many of the statistical analysis methods of other structural equation studies. If you tell us about a specific additional statistical analysis method, we will add it.

In the discussion section, please do not repeat results section while summarise research findings.

Response: We appreciate your valuable comments. As suggested, we revised the repetitive part and added summary research findings.

I think you should add the Conclusion part. Please revise it.

Response: We appreciate your valuable comments. As suggested, we added a Conclusion part.

Reviewer 2 Report

This is a unique and well-developed study. Purpose of the study is clearly identified in the introduction and previous studies are well reviewed related to the constructs used in this study. However, there are couple suggestions to improve the manuscript.

1. Method
The items seemed to have been just taken from the literature, with little adaptation to the actual context of the study.

This poses certain problems, as the study remains a replication of previous research.

Was non-bias analyzed? How? Please show.

2. Implications

Theoretical contribution should be more elaborated in the discussion section.

Author Response

We appreciate the opportunity to revise and resubmit our paper titled “Structural Relationships among Strategic Experiential Modules, Motivation, Serious Leisure, Satisfaction and Quality of Life in Bicycle Tourism.” We are grateful for the time you put forth for your critiques and revision recommendations. Your vital comments and suggestions have contributed substantially to improving the presentation and overall quality of our study. The corresponding changes and refinements made in the revised paper are summarized in our responses below.

Method
The items seemed to have been just taken from the literature, with little adaptation to the actual context of the study. This poses certain problems, as the study remains a replication of previous research. Was non-bias analyzed? How? Please show.

Response: We appreciate your valuable comments. As suggested, we think the reviewer's opinion is very reasonable. In order to further enhance the uniqueness of the paper, a conclusion part has been added and the introduction and conclusion have been rewritten. The content of non-bias is not fully understood, so please tell us more specific details, and we will revise accordingly.

Implications

Theoretical contribution should be more elaborated in the discussion section.

Response: We appreciate your valuable comments. As suggested, based on the reviewers' comments, theoretical contribution has been rewritten in the conclusion section.

Also missing is literature related to the case study such as “How to develop a sustainable and responsible hallmark sporting event? - Experiences from tour of Qinghai Lake International Road Cycling Race, using IPA method”.

Response: We appreciate your valuable comments. As suggested, we have added the relevant pre-research section

Reviewer 3 Report

Structural relationships among strategic experiential modules, motivation, serious leisure, satisfaction and quality of life in bicycle tourism

The topic is interesting, but the reliability of the research, in terms of reproducibility and representativeness of the sample, is doubtful. In addition, it is necessary to address some modifications to improve the information offered to readers.

Abstract

“in order to encourage more bicycle tourists to visit Qinghai Lake”

While a scientific article may have managerial and practical implications, it cannot be approached as a marketing strategy for a particular tourist destination.

Keywords: Three to ten pertinent keywords need to be added after the abstract. We recommend that the keywords are specific to the article, yet reasonably common within the subject discipline.

Some keywords are missing such as: strategic experiential modules; structural equation modeling; Qinghai Lake; China

Materials and Methods

Readers need to know the context of the case study, so you should add a subsection with this information. On the one hand, COVID-19 has been a before and after in the tourism industry worldwide. Therefore, it is necessary to mention that the data and findings predate the pandemic outbreak. On the other hand, there is a lack of official statistics on the people who regularly participate in the race and the tourist influx in the area. Looking at the origin of tourists, are the visitors domestic or foreigners? Also missing is literature related to the case study such as “How to develop a sustainable and responsible hallmark sporting event? - Experiences from tour of Qinghai Lake International Road Cycling Race, using IPA method”.

“Using random sampling...”

The representativeness of the sample is not clear, because readers are unaware of the volume and composition of the target population, and the majority of respondents are young men living in China. These circumstances should be reflected in the subsection of Limitations / Future work.

Discussion: Authors should discuss the results and how they can be interpreted in perspective of previous studies and of the working hypotheses. The findings and their implications should be discussed in the broadest context possible and limitations of the work highlighted. Future research directions may also be mentioned.

Conclusions: This section is mandatory, and should provide readers with a brief summary of the main conclusions (please see instructions for authors).

Appendix A

In Table A1, a column is missing with the references of the literature related to the constructs and items.

References:

Please check the citations and references: Multidisciplinary Digital Publishing Institute (MDPI) style. I use Mendeley.com with the MDPI style.

Author Response

We appreciate the opportunity to revise and resubmit our paper titled “Structural Relationships among Strategic Experiential Modules, Motivation, Serious Leisure, Satisfaction and Quality of Life in Bicycle Tourism.” We are grateful for the time you put forth for your critiques and revision recommendations. Your vital comments and suggestions have contributed substantially to improving the presentation and overall quality of our study. The corresponding changes and refinements made in the revised paper are summarized in our responses below.

Abstract

“in order to encourage more bicycle tourists to visit Qinghai Lake”

While a scientific article may have managerial and practical implications, it cannot be approached as a marketing strategy for a particular tourist destination. 

Response: We appreciate your valuable comments. As suggested, we re and revised this section

Keywords: Three to ten pertinent keywords need to be added after the abstract. We recommend that the keywords are specific to the article, yet reasonably common within the subject discipline. Some keywords are missing such as: strategic experiential modules; structural equation modeling; Qinghai Lake; China

Response: Thank you very much for your valuable suggestions. As suggested, we revised this sentence as follow

Keywords: bicycle tourism; motivation; serious leisure; satisfaction; quality of lifeï¼›structural equation modeling; Qinghai Lake; China

Readers need to know the context of the case study, so you should add a subsection with this information. On the one hand, COVID-19 has been a before and after in the tourism industry worldwide. Therefore, it is necessary to mention that the data and findings predate the pandemic outbreak. On the other hand, there is a lack of official statistics on the people who regularly participate in the race and the tourist influx in the area. Looking at the origin of tourists, are the visitors domestic or foreigners?

Response: Thank you very much for your valuable suggestions. As suggested, we have added explanations to the description and results of the questionnaire, allowing readers to more directly understand the source and background of the data

“Using random sampling...”

The representativeness of the sample is not clear, because readers are unaware of the volume and composition of the target population, and the majority of respondents are young men living in China. These circumstances should be reflected in the subsection of Limitations / Future work

Response: Thank you very much for your valuable suggestions. As suggested, we revised and elaborated the subsection of Limitations / Future work

Discussion: Authors should discuss the results and how they can be interpreted in perspective of previous studies and of the working hypotheses. The findings and their implications should be discussed in the broadest context possible and limitations of the work highlighted. Future research directions may also be mentioned.

Response: Thank you very much for your valuable suggestions. As suggested,
we added a description of the limitations of the research and future research directions.

Conclusions: This section is mandatory, and should provide readers with a brief summary of the main conclusions (please see instructions for authors).

Response: Thank you very much for your valuable suggestions. As suggested,
We have added a conclusion section to the main text.

Appendix A

In Table A1, a column is missing with the references of the literature related to the constructs and items.

Response: Thank you very much for your valuable suggestions. As suggested, We have added an explanation of Table A1 in the text.

References:

Please check the citations and references: Multidisciplinary Digital Publishing Institute (MDPI) style. I use Mendeley.com with the MDPI style.

Response: Thank you very much for your valuable suggestions. As suggested, we have revised this part strictly in accordance with the MDPI style  requirements

Round 2

Reviewer 3 Report

Structural relationships among strategic experiential modules, motivation, serious leisure, satisfaction and quality of life in bicycle tourism

The article has improved significantly and I therefore recommend its publication in IJERPH, after addressing some minor changes.

Appendix A

In Table A1, a column (Related literature), with the references of the literature related to the constructs and items, is missing.

References:

Please capitalise all major words in the journal title; e.g: Annals of Tourism Research

Good luck!

Author Response

We appreciate the opportunity to revise and resubmit our paper titled “Structural Relationships among Strategic Experiential Modules, Motivation, Serious Leisure, Satisfaction and Quality of Life in Bicycle Tourism.” We are grateful for the time you put forth for your critiques and revision recommendations. Your vital comments and suggestions have contributed substantially to improving the presentation and overall quality of our study. The corresponding changes and refinements made in the revised paper are summarized in our responses below.

Appendix A

In Table A1, a column (Related literature), with the references of the literature related to the constructs and items, is missing.

Response: We appreciate your valuable comments. As suggested, we have modified the Table A1: Related Literature section.

References:

Please capitalise all major words in the journal title; e.g: Annals of Tourism Research

Response: We appreciate your valuable comments. As suggested, we revised the References section.